# Electrically Induced Structural Transformations of a Chiral Nematic under Tangential-Conical Boundary Conditions

**DOI:** 10.3390/molecules28237842

**Published:** 2023-11-29

**Authors:** Denis A. Kostikov, Mikhail N. Krakhalev, Oxana O. Prishchepa, Victor Ya. Zyryanov

**Affiliations:** 1Kirensky Institute of Physics, Federal Research Center KSC SB RAS, Krasnoyarsk 660036, Russia; kmn@iph.krasn.ru (M.N.K.); p_oksana@iph.krasn.ru (O.O.P.); zyr@iph.krasn.ru (V.Y.Z.); 2Institute of Engineering Physics and Radio Electronics, Siberian Federal University, Krasnoyarsk 660041, Russia

**Keywords:** chiral nematic, cholesteric, conical anchoring, polymethacrylate, director tilt angle, electrically induced transformation, orientational structure, polarizing optical microscopy

## Abstract

In this study, structural transformations induced by an electric field in the chiral nematic under tangential-conical boundary conditions have been considered. The composition influence of the orienting polymer films on the director tilt angles, the formation of orientational structures in the LC layer, as well as the electro-optical response and relaxation processes have been studied. It has been shown that the poly(tert-butyl methacrylate) concentration change in the orienting polymer mixture allows for smoothly controlling the director tilt angle without fixing its azimuthal orientation rigidly.

## 1. Introduction

Liquid crystals (LC) are materials of great interest, owing to their anisotropy of physical properties and specific orientational molecular ordering [1,2]. The unique electro-optical characteristics of LCs underlie their application in various optoelectronic devices, such as displays [3], controllable lasers [4,5], LC lenses [6], diffraction gratings [7,8], smart windows and glasses [9,10], etc. [11,12].

The orientational LC structure is determined by some factors, including the boundary conditions, which are set by thin orienting films for LC layers. The boundary conditions specify the orientation of the director **n** (the unit vector directed along the preferred orientation of molecule long axes) at the interface. There are four types of boundary conditions [13]:The boundary conditions, under which the director is in the orienting film plane and has a zero pretilt angle, are identified as the tangential $\theta_{T}=0^{\circ }$;The boundary conditions, under which the director is perpendicular to the orienting film plane, are identified as homeotropic $\theta_{H}=90^{\circ }$;The tilted boundary conditions are characterized by the fixed azimuthal director orientation and the tilt angle $\theta_{Ti}$, differing from $0^{\circ }$ and $90^{\circ }$;The boundary conditions, under which the tilt angle $0^{\circ }<\theta_{C}<90^{\circ }$ and the director orientation has azimuthal degeneration, are identified as conical.

The director orientation under conical boundary conditions is characterized by the fixed polar angle at the interface and a possibility of azimuthal gliding creating the cone of revolution [14]. The search for the polymer compositions causing the easier azimuthal turn of the director is an urgent problem for LC researchers. The application of such materials will allow the observation of the unconventional orientation-structural effects, obtaining an atypical response on the external action as well as reducing the operating voltage in LC devices. At present, such compositions have not been studied enough. Particularly, it is caused by a choice complexity of an orienting coating that provides the conical anchoring. The conical boundary conditions with the polar tilt angle specified by the chemical properties of LC at the interface with its own isotropic phase have been considered in [15]. To form the conical anchoring it can add the colloidal plates into LC [16] or apply on substrates the special compounds [17]. In practice, the orienting coatings based on polymers are more suitable. In this case, the setting of conical boundary conditions does not require the special techniques for the polymer film processing. There are some polymers providing the conical anchoring for LC. For instance, the polystyrene and polyisoprene assign the conical anchoring for nematic 5CB with the tilt angle $\theta_{C}=18^{\circ }$ and $\theta_{C}=16^{\circ }$, respectively [14]. The perfluoropolymer sets for LC CCN-47 the conical anchoring with the $\theta_{C}$ angle depending on temperature [18]; the poly(isobutyl methacrylate) (PiBMA) assigns the conical anchoring for the nematic LN-396 with $\theta_{C}=47.7^{\circ }$ [19].

Chiral-nematic (cholesteric) liquid crystals (CLC) have a helicoidal director structure [20]. Owing to this peculiarity, another parameter, in spite of boundary conditions, specifies that the orientational ordering is the ratio of the helix pith *p* (the distance by which the director turns by 2π) to the LC layer thickness *d*. Under the tangential-conical boundary conditions, the homogeneous azimuthal orientation of the director is specified by the rubbing direction at the substrate with tangential anchoring, while it is set by the ratio d/p at the substrate with conical anchoring [21]. Such a combination of surface anchoring opens up new possibilities of CLC application in the electro-optical devices; it can also promote the operating characteristics of the already available LC materials. The conical anchoring characterized by an easy director gliding (the possibility of director azimuthal rotation on the substrate with conical anchoring) has two key advantages: (i) the non-threshold response of the structure to an electric field is applied both perpendicularly and along the substrates; (ii) it realizes the smooth control of both the polar and director azimuthal angle with an electric field, a change in d/p, and other factors.

For example, the formation of the tangential-conical boundary conditions due to the wetting effect allows one to create the periodical structure whose orientation is determined by the cholesteric layer thickness in the mesophase [22,23]. In turn, the mesophase layer thickness is controlled by a change in temperature near the point of LC phase transition. Under the tangential-conical boundary conditions assigned by a polymer, the various electro-optical effects are possible. So, it can create the electrically controlled diffraction grating, the orientation of which depends on the voltage value applied during the periodic structure formation [24]. After that, the periodic structure orientation remains after switching off the voltage. In the case of the defect-free orientational structure arising under the tangential-conical anchoring, the linear polarization of the radiation passed through the LC layer can be controlled [25]. Here, the azimuthal director orientation at the surface with conical anchoring and, accordingly, the azimuth of the linear polarization of transmitted light depends on the voltage value applied to the CLC layer. A sensitivity of the azimuthal director orientation at the substrate with conical anchoring to the d/p ratio was used in [26] to realize the complete control of polarization parameters of a light passed through the cholesteric. In this case, the photosensitive cholesteric with the helix pitch varying under radiation was used. The polarization azimuth of passed light was assigned with the helix pitch *p*, and the polarization ellipticity was controlled by a low electric field influencing the effective anisotropy of the refractive index $\Delta n_{eff}$ of the LC layer.

The above-described examples have demonstrated two main advantages of conical anchoring characterizing a reaction of cholesteric structure under tangential-conical boundary conditions. Thus, the value of the polar tilt angle $\theta_{C}$ of the director at the substrate with conical anchoring must influence the structure response and its optical parameters, e.g., the effective anisotropy of the refractive index. Consequently, a variation ability of the $\theta_{C}$ angle is useful to optimize the LC system. Moreover, a change in the $\theta_{C}$ angle by any method should not result in the critical degradation of a director capacity to gliding. Earlier, we demonstrated the possibility to specify the different director tilt angles on the substrate with conical anchoring using a mixture of polymethacrylates [19]. In the present work, the electrically induced structural transformations within CLC cells with tangential-conical boundary conditions at the various director tilt angles on the substrate under conical anchoring are considered.

## 2. Results

As was mentioned above, the PiBMA polymer assigns a conical anchoring for the nematic LN-396 with the director tilt angle $\theta_{C}=47.7^{\circ }$. The addition of poly(methyl methacrylate) (PMMA) or poly(tertbutyl methacrylate) (PtBMA) to PiBMA results in a decrease in the angle $\theta_{C}$ as the polymer’s weight content increases. If the PiBMA content in the orienting PiBMA: PMMA mixture is above 40%, then on the substrate with conical anchoring, the CLC initially contains domain structures with tilt angles of the same magnitude but different signs. The domains are divided from each other by the clearly marked interface (the domain border), where the boundary conditions at the substrate with conical anchoring are violated. Therefore, the domain border has the topological surface linear defect located at the substrate with conical anchoring. Near this linear defect, the additional twist deformation of the director arises in such a manner that the director at the substrate with conical anchoring in the domain border center is parallel to the linear defect [19,21].

To determine the azimuthal director orientation on the substrate with conical anchoring, the rotating analyzer method can be applied for the defect-free structure and the structure containing linear defects, or the periodic structure (Figure 1):The sample is placed on the microscope stage so that it is illuminated from the side of the substrate with tangential anchoring;The rubbing direction **R** of the substrate with tangential anchoring is oriented perpendicularly to the polarizer direction *P*;The analyzer *A* is rotated into a position corresponding to the intensity minimum of passed light for the considered sample area;In this case, according to the waveguide Mauguin regime, the orientation of analyzer *A* is parallel to the azimuthal director orientation at the substrate with conical anchoring [27].

Studying the director distribution near the linear defects (domain borders) by the rotating analyzer method, it is necessary to focus on them (i.e., to observe the substrate with conical anchoring). Then, a series of photos should be taken at different orientations of the analyzer in the range from 0 to $180^{\circ }$ with the fixed step, e.g., $10^{\circ }$. The azimuthal director orientation near the defects is determined by the position of the extinction bands observed for each analyzer *A* orientation.

The dielectric anisotropy of the used LC is positive, and the director tends to align along the field direction under a voltage (it is perpendicular to substrate plane). At a relatively low voltage (up to 0.5 V), no noticeable orientational transformations were observed. Further, as the voltage increases, the director turns azimuthally near the substrate with the conical anchoring, which is revealed as a change in the orientation of the linear polarization of light passed through the LC layer. The observable reduction in the azimuthal twist angle $\phi_{dir}$ of the structure as the applied voltage increases is conditioned by two factors: (i) a growth of the director tilt angle θ in LC bulk; (ii) a subsequent growth of the helix pitch *p* of cholesteric. This change in the azimuthal director orientation can be considered as the reason for the decrease in the effective value of the d/p ratio. As a result, the domain structure becomes unstable and leads to a gradual decrease in the domain sizes accompanied by a movement in their borders [27]. Finally, the domain characterized by the opposite director tilt angle collapses fully, and the domain border annihilated.

A similar transformation was observed for PiBMA:PtBMA polymer mixture as the orienting film (where PtBMA is the poly(tertbutyl methacrylate)); however, the domain structure appears at the PiBMA concentration of more than 30%. Below, the peculiarities of the electrically induced response of the structures formed under the tangential-conical anchoring specified by PiBMA: PMMA (where PMMA is poly(methyl methacrylate)) and PiBMA:PtBMA were considered in detail.

### 2.1. LC Cells Based on PiBMA:PMMA Films

Figure 2a demonstrates the initial texture within the CLC cell with the PiBMA:PMMA:LN-396 = 80:20:20 film before applying the voltage and the director tilt angle $\theta_{C}=36.7^{\circ }$. At U=10 V, the domains’ size decreases slowly and their borders move (Figure 2b,c), resulting in the appearance of the areas with a different azimuthal orientation and transienting smoothly into the homogeneous state for hours. Such a temporary memory-like effect of the azimuthal director orientation was observed in the nematic with similar orienting films [19].

The initial domain structure does not restore after switching off the voltage at the considered *d* and d/p=0.6 [27]. Moreover, the appeared structure with the inhomogeneous azimuthal orientation of the director relaxes into the homogeneous-twisted state, this process proceeds unevenly. The director turns more actively for the first few minutes after switching off the electric field (Figure 2d,f,g). Further, the relaxation is revealed only in a minor twist of the director. Figure 2h shows the azimuthal director orientation in the area marked in Figure 2d, obtained with the rotating analyzer method. The director turns approximately $\pm 144^{\circ }$ (from the dark area to a bright area in Figure 2d) within the area with strong deformation. This transition is accompanied by a sharp change in the azimuthal director orientation.

The structure under study relaxes into the azimuthally uniform state for approximately 300 min (Figure 2g). It should be emphasized that the process continues from approximately ten to hundreds of minutes subject to the exposure time, the electric field amplitude, the director tilt angle at the conical interface and the initial domain size. For example, the smaller domains appear, and the shrinking and relaxation into the homogeneous director orientation occurs faster as the director tilt angle decreases. On the other hand, the relaxation time rises as the domain shrinking time increases, for example, due to a reduction in the amplitude of the applied voltage. The similar gliding effects of the director in the azimuthal plane subject to the field amplitude and its duration time were observed in nematics [28]. The temporary memory-like effect arising at the decrease in domain sizes is obviously conditioned by a breaking of conical anchoring at the border between domains. Accordingly, the longer the boundary conditions are violated in any sample area (i.e., the slower the domain border moves), the longer the relaxation time of the temporary structure into a stable state.

In addition, the above described scenario of structure transformation is observed in the cells with the orienting PiBMA:PMMA:LN-396 films at a concentration of PMMA ≤60% with a switched on and off voltage. Here, the structures with strong (or noticeable) director deformations appear under a domain shrinking in all cells. Usually, the cell with the wide conical angle of the director tilt (at high PiBMA concentration) relaxes faster than the cell with the smaller tilt angle $\theta_{C}$. The stated effect can be explained by an increase in PMMA concentration that, apparently, promotes the smaller azimuthal director mobility at the interface. The samples with the tilt angle $\theta_{C}<10^{\circ }$ forming at the PMMA concentration more than 60% in the orienting polymer PiBMA:PMMA:LN-396 mixture show the more complex response to the electric filled, and they are considered in Section 2.4. In all cases, the temporary memory-like structures resulting from the switched-off voltage are characterized only by the azimuthal deviation of the director orientation from the rubbing direction.

### 2.2. LC Cells Based on PiBMA:PtBMA Films

The samples based on the PiBMA:PtBMA polymer mixtures demonstrate another transformation type. Figure 3a shows the initial domain texture of the CLC layer with the orienting film PiBMA:PtBMA:LN-396 = 60:40:20 specifying the director tilt angle $\theta_{C}=36.4^{\circ }$. The homogeneous azimuthal orientation of the director appears at the shrinking of domain borders under a voltage (Figure 3b,c), and it does not show the memory-like structure. Therefore, the structure relaxes into the state with the homogeneous azimuthal orientation of the director on the substrate with conical anchoring for a few seconds when the voltage is switched off (Figure 3d). The homogeneous texture of the director characterized by the minimal light transmission in Figure 3d corresponds to the twist angle $\phi_{dir}=-198^{\circ }$.

Such a transformation type is observed for all considered samples based on the polymer mixtures PiBMA:PtBMA. The structure relaxes to the homogeneous state between one and several tens of a second in the samples with PiBMA ≥30%. The defect-free homogeneous structure is observed initially at a low PiBMA concentration (<30%), when the near-zero director tilt angle is formed at the conical interface ($\theta_{C}≅$ 1–2${}^{\circ }$). A destruction of this structure under the voltage applied to the sample is described in Section 2.4.

### 2.3. Azimuthal Rotation of Linear Polarization of Light Passed through CLC Layer

The inhomogeneous director configuration in the samples based on PMMA relaxes for a long time after switching off the field. The response and relaxation processes occur sufficiently fast in the sample based on PtBMA; thus, the transient storage of the azimuthal director orientation is not observed. Thus, it is revealed in the response of the defect-free structure to an electric field. Figure 4a,e present the initial textures of the CLC cell with the orienting films based on PiBMA:PMMA:LN-396 = 80:20:20 and PiBMA:PtBMA:LN-396 = 60:40:20, for which there is an azimuthal twist angle of the structure $\phi_{dir}=-202^{\circ }$ and $\phi_{dir}=-198^{\circ }$ in voltage-off, respectively. A weak change in the director twist angle is observed in CLC cells with the orienting film based on PMMA depending on the voltage magnitude. It decreases up to $\phi_{dir}=-178^{\circ }$ at U=1.4 V (Figure 4a–d). At the same time, the stronger change in the director twist angle under the voltage is observed in the CLC cell with the orienting film based on PtBMA. Here, the twist angle decreases up to $\phi_{dir}=-131^{\circ }$ at U=1.4 V (Figure 4e–h). Therefore, the limit of the change in the light polarization turning angle for PiBMA:PMMA:LN-396 = 80:20:20 and PiBMA:PtBMA:LN-396 = 60:40:20 films, for which the Maugin regime still fulfills at the white light, is $21^{\circ }$ and $66^{\circ }$, respectively. This means that the structure behavior is conditioned for both the director tilt angle at the conical interface and the interaction type of the polymer mixture in the orienting films. These factors also affect the process dynamics of the structure response. A further voltage increase results in a breakdown of the waveguide regime, which is revealed in the intensity rising of the transmitted light corresponding to the minimum transmittance for this measurement scheme (Figure 4d,h).

The PtBMA polymer stimulates the larger turn of the azimuthal orientation of polarization in comparison with PMMA, owing to the better director gliding at the conical anchoring interface. Thus, this material is more promising for its application.

### 2.4. Inhomogeneous Azimuthal Transformation of the Director

As the portion of the PMMA or PtBMA polymer assigning the conical anchoring increases, the director tilt angle decreases; then, the degenerate state of the azimuthal director orientation remains [19]. As a result, the domains’ size decreases and the rate of their shrinking increases. At PiBMA:PMMA:LN-396 = 30:70:20 and PiBMA:PtBMA:LN-396 = 20:80:20, the defect-free twisted structure is formed initially. Figure 5a shows the cell example with the director tilt angle $\theta_{C}≅1^{\circ }$, which the orienting film PiBMA:PtBMA:LN-396 = 20:80:20 provides. Obviously, the structure is initially azimuthally homogeneous. This does not change practically under voltage U≤0.6 V (Figure 5b). The structure undulations (Figure 5c) characterized by the periodical distribution of the azimuthal director orientation appear at voltage U=1.2 V for 3 min. Further electric action of the same magnitude amplifies the undulation amplitude (Figure 5d–f) and the optical texture of stripes is more bright. Such transformations can be explained by the appearance of periodic director configurations within the LC bulk observable in the twisted structures under small director tilt angles (less than $5^{\circ }$) [29], which lead to the periodic distribution of the azimuthal director orientation at the substrate with conical anchoring. The structure returns to its initial state entirely after switching off the electric field during 40 min (Figure 5g). The azimuthal director orientation at the substrate with conical anchoring in the area marked in Figure 5d is presented schematically in Figure 5h. The director distribution was identified by the rotating analyzer method. Considering the director field in the direction from the red line to the blue one or vice versa (both lines are marked in the Figure 5d), one can notice an azimuthal turn by $\pm 120^{\circ }$. Thus, the observed undulations are characterized by the periodic change in the $\phi_{dir}$ angle in the range from $-130^{\circ }$ to $-250^{\circ }$. The undulation period is about 70 μm. It should be emphasized that the relaxation time and the director distribution depend both on the electric field duration and its magnitude.

The complicated director reorientation under the voltage at the tilt angle $\theta_{C}<10^{\circ }$ increases the relaxation time significantly, even in the case of orienting PtBMA films providing better director gliding in comparison with PMMA films. The relaxation processes in the samples based on PMMA in this case are also slower, and the undulation traces caused by the memory-like structures can be meta-stable for tens of hours.

## 3. Discussion

In this paper, the electrically induced response and relaxation processes of the chiral nematic at tangential-conical boundary conditions have been studied. An influence of the orienting film composition on the director tilt angle $\theta_{C}$ has been shown, in addition to the orientational structures and the response and relaxation processes after switching the on/off voltage. As was found, a transient azimuthal inhomogeneity is observed within the structure under the voltage, independently of the PMMA content in the orienting PiBMA: PMMA films. It persists from ten to several hundred minutes. The memory-like structures are not formed at the PiBMA:PtBMA orienting coating, and the director structure relaxes within several tens of a second after switching-off the field. Consequently, the director is twisted by the larger angle using the polymer mixture with PtBMA in comparison with PMMA. This effect can be explained by the easier gliding of the director on the polymer PiBMA:PtBMA film, which is comparable to the director gliding on the pure PiBMA film. PMMA added to the orienting film decreases the rate of the director gliding significantly, which results in an increase in the relaxation time of the structure by the orders of magnitude. It has been demonstrated that the defect-free configuration is formed initially under the tangential-conical anchoring at $\theta_{C}<10^{\circ }$, which is transformed into the structure with a periodic azimuthal changing of the director orientation under the voltage.

The PtBMA-based cells under study, with the smoothly operated director tilt angle due to the conical boundary conditions, can improve the parameters of present devices as well as develop novel ones. For instance, an application of the conical boundary conditions with different tilt angles provides a continuous tuning of the parameters of diffraction LC gratings [24] or the electrically controlled rotator of the light polarization based on CLC with tangential-conical anchoring [25]. The obtained results can be used to develop the polarization transformers, the tunable phase plates and other electro-optical devices.

The observed memory-like structures in the samples based on PMMA films can be considered as the metastable states, the lifetime of which is determined by the composition of the orienting film and the value of the applied voltage. At that, the metastable structures appearing after a domain shrinking are nonrenewable, unlike the undulations. The application of two- or many-component orienting films allows for fine-tuning the balance between the switching parameters (the operating voltage, the exposure time, etc.) of undulations in a metastable state and their lifetime. These memory-like structures are of interest in order to develop the electro-optical devices with reduced operating voltages, switchable diffraction gratings, etc.

## 4. Materials and Methods

### 4.1. Materials

The sandwich-like CLC cells consisting of two glass substrates with the ITO (indium tin oxide) coating were studied. The orienting polymer film was deposited over the ITO layer. Polyvinyl alcohol (PVA) (Sigma Aldrich, St. Louis, MO, USA) was used as the orienting polymer with tangential anchoring. The conical boundary conditions were set by the mixture of poly(isobutyl methacrylate) (PiBMA) (Sigma Aldrich, St. Louis, MO, USA) and poly(methyl methacrylate) (PMMA) (Sigma Aldrich, St. Louis, MO, USA), or PiBMA and poly(tertbutyl methacrylate) (PtBMA) (Sigma Aldrich, St. Louis, MO, USA). Nematic LC LN-396 (Belarusian State Technological University, Minsk, Belarus ), at a concentration of 20% of the polymer weight, was added to the polymethacrylates to provide a homogeneity of the LC structure and to eliminate an azimuthal inhomogeneity of the director orientation in CLC cells caused by a flow effect [19]. The melting temperature of LN-396 is $T_{m}=-20{\;}^{\circ }$C, the clearing point is $T_{c}=+66{\;}^{\circ }$C, and the dielectric anisotropy is Δϵ=10.2. The weight ratios of polymers and the nematic in the orienting films, namely PiBMA:PMMA:LN-396 or PiBMA:PtBMA:LN-396, were ranged from 90:10:20 to 60:40:20.

### 4.2. Sample Preparation

The orienting films were deposited by the spin coating, and thereafter they were dried for 1 h at 60 ${}^{\circ }$C temperature. Then, the PVA film was unidirectionally rubbed by Rubbing Machine HO-IAD-BTR-01 (Holmarc, India) to specify the homogeneous boundary conditions. The films based on PiBMA: PMMA and PiBMA:PtBMA were not treated after drying. The substrates were glued by the photosensitive compound NOA61 (Norland Products, Cranbury, NJ, USA) using the glass microspheres of the 17.3±1.4 μm diameter (Duke Scientific, Palo Alto, CA, USA). The LC layer thickness was measured by the interference method before LC filling. Next, the cells were filled with the nematic LN-396 doped with the chiral additive cholesteryl acetate (X3) (Sigma Aldrich) at room temperature with capillary force. The helical twisting power of the cholesteryl acetate in LN-396 is β=6.94μm${}^{-1}$, and its concentration was chosen particularly for every sample to provide d/p=0.6. Such a d/p ratio, as a rule, is characterized by the twist angle of the structure $\phi_{dir}≅200^{\circ }$ at the director tilt angle $\theta_{C}≅50^{\circ }$. Consequently, the conditions of the waveguide Mauguin regime are satisfied at the cholesteric layer thickness d≅20μm.

### 4.3. Measurements and Processing

The director tilt angles set with polymethacrylate mixtures were determined by the light transmittance dependence of nematic LC cells on the incidence angle [19,30]. Such dependence was recorded with the optical setup consisting of a He-Ne laser (λ=632.8 nm), polarizer, rotating platform with LC cell, analyzer and photo-detector. The measurements were carried out with the pure nematic LN-396, since the method requires the unidirectionally azimuthal director orientation throughout the LC bulk.

The orientational CLC structures and their transformations under an electric field were investigated by the polarizing optical microscopy (POM) with the Axio Imager.A1m (Zeiss, Germany) microscope. The AC voltage of varied amplitudes at 1 kHz was applied perpendicular to the LC layer.

## Figures and Tables

**Figure 1 molecules-28-07842-f001:**
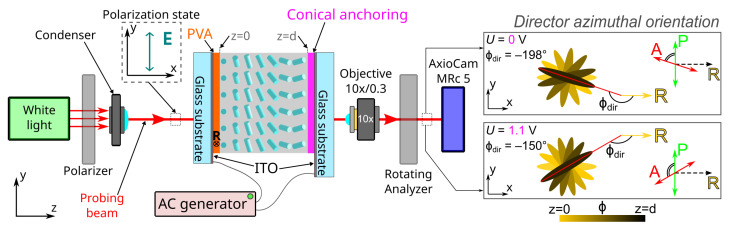
Optical scheme to determine the azimuthal director orientation at the substrate with conical anchoring by the rotating analyzer method. The azimuthal director orientation is characterized by ϕ(z) angle, ϕ(z)=0 at z=0 (the border of LC layer with tangential anchoring), and $\phi \left(z\right)=\phi_{dir}$ at z=d (the border of LC layer with conical anchoring). The insertion on the right shows the azimuthal director orientation varying along *z*-axis schematically for the angles $\phi_{dir}=-198^{\circ }$ (the angle between analyzer and polarizer is $72^{\circ }$) and $\phi_{dir}=-150^{\circ }$ (the angle between analyzer and polarizer is $120^{\circ }$), as well as the corresponding orientations of polarizer *P* and analyzer *A* when the minimal light transmission is observed.

**Figure 2 molecules-28-07842-f002:**
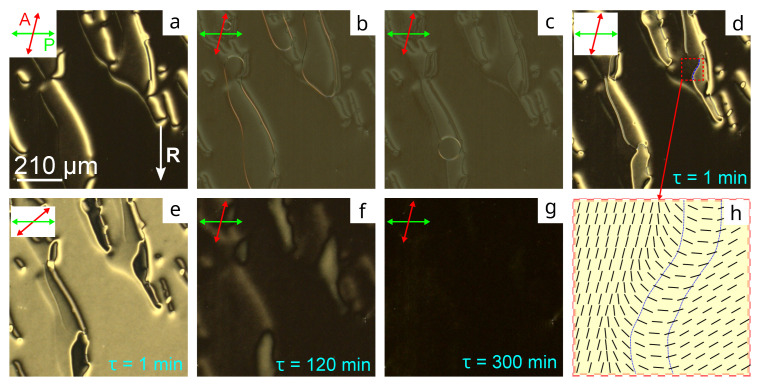
POM photos of CLC cell area with the orienting film based on PiBMA:PMMA:LN-396 = 80:20:20 mixture taken in (**a**) initial state, (**b**) 1 min and (**c**) 10 min after voltage—on at U=10 V, and (**d**,**e**) 1 min, (**f**) 120 min, (**g**) 300 min after voltage—off. Here and below, the vector **R** shows the rubbing direction on the substrate with tangential anchoring, and the double arrows indicate the orientation of polarizer and analyzer. (**h**) Scheme of the azimuthal director orientation at the substrate covered with PiBMA:PMMA:LN-396 = 80:20:20 mixture 1 min after voltage-off. Photos are taken at the angle between analyzer and polarizer (**a**–**d**,**f**,**g**) $76^{\circ }$, (**e**) $40^{\circ }$.

**Figure 3 molecules-28-07842-f003:**
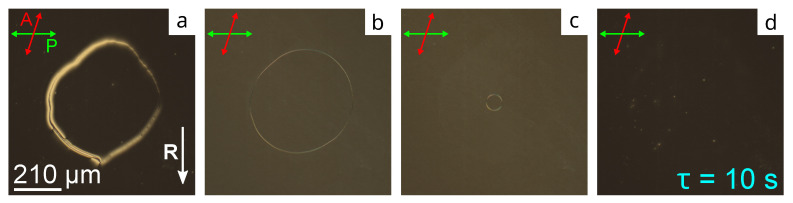
POM photos of CLC cell area with the orienting film based on PiBMA:PtBMA:LN-396 = 60:40:20 mixture. The texture (**a**) before, (**b**) in 3 min, (**c**) in 27 min after voltage-on at U=10 V, and (**d**) in 10 s after voltage-off. Photos are taken at the angle $72^{\circ }$ between analyzer and polarizer.

**Figure 4 molecules-28-07842-f004:**
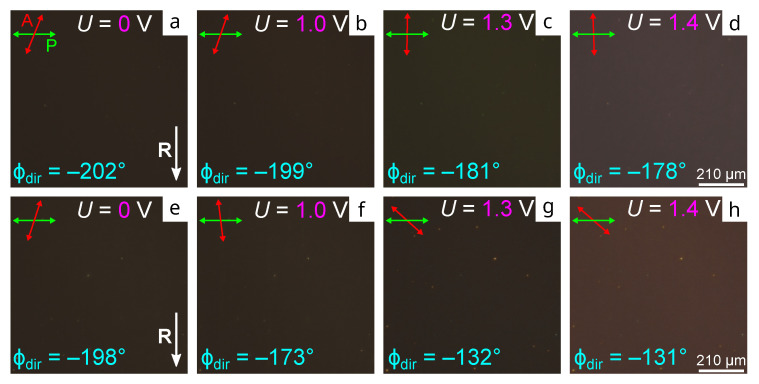
POM photos of CLC cells with the orienting films based on the polymer mixture and LC with the weight ratio (**a**–**d**) PiBMA:PMMA:LN-396 = 80:20:20 and (**e**–**h**) PiBMA:PtBMA:LN-396 = 60:40:20. (**a**,**e**) The area before, and (**b**,**f**) in voltage-on at U=1.0 V, (**c**,**g**) U=1.3 V, and (**d**,**h**) U=1.4 V.

**Figure 5 molecules-28-07842-f005:**
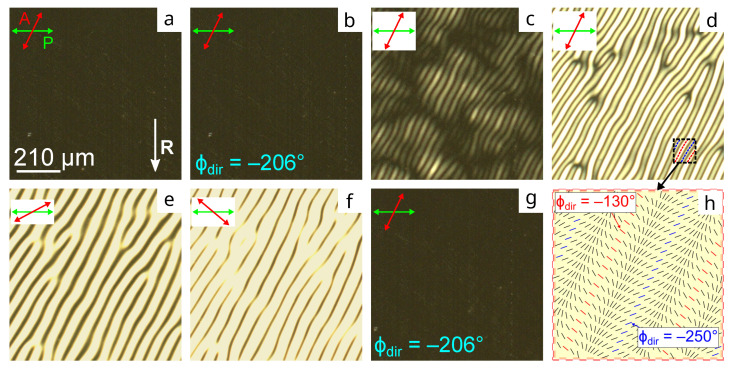
POM photos of CLC cell with the orienting film based on PiBMA:PtBMA:LN-396 = 20:80:20 mixture. (**a**) Initial texture, (**b**) in voltage-on at U=0.6 V, (**c**) in 3 min, (**d**–**f**) 7 min after voltage-on at U=1.2 V, and (**g**) 40 min after voltage-off. (**h**) Scheme of the azimuthal director orientation at the substrate covered with PiBMA:PtBMA:LN-396 = 20:80:20 mixture in 7 min after voltage-on at U=1.2 V. Photos are taken at the angle between analyzer and polarizer (**a**–**d**) and (**g**) $64^{\circ }$, (**e**) $20^{\circ }$, and (**f**) $140^{\circ }$.

## Data Availability

The data presented in this study are available on request from the corresponding author.

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
