# Peer review of "Electrically Induced Structural Transformations of a Chiral Nematic under Tangential-Conical Boundary Conditions"

_molecules, 2023, doi:10.3390/molecules28237842_

Round 1

Reviewer 1 Report

Comments and Suggestions for Authors

In this manuscript, the authors investigate the electrically induced optical response and relaxation processes of cholesteric liquid crystals (CLC) under tangential-conical boundary conditions. Two different orienting polymer mixtures, PiBMA:PMMA and PiBMA:PtBMA, are used to achieve conical boundary conditions, and their effects on the CLC textures under external electric fields are compared. The defect lines, relaxation processes, and changes in the azimuthal orientation angle of the liquid crystal director are analyzed. The manuscript has potential for publication, but I recommend addressing the above issues to improve the clarity, completeness, and significance of the work.

I have specific questions and comments that I would like the authors to address:

Ÿ   Insufficient background information: The introduction should provide more context on the importance and applications of tangential-conical boundary conditions in CLC systems. Without a clear understanding of the significance of such complex orientations, readers may not appreciate the importance of this work.

Ÿ   Although the authors have successfully demonstrated the ability to specify different tilt angles in a nematic liquid crystal (NLC) system (Ref. 18), I would like the authors to explain why they chose to study the structural transformations in a CLC system for this study. The helical twisting energy of CLC and the anchoring of the tilt angle appear to introduce complex interactions into the system.

Ÿ   The four figures mainly consist of POM images, which provide a representation of the experimental results. However, additional schematic diagrams could be included or provide characterization results using other methods.

Ÿ   Could the authors clarify the method used for the reconstructed azimuthal director orientation shown in Figure 1h? Is it based on polarized optical microscopy (POM) images?

Ÿ   I believe the quality of the article could be further improved by discussing the potential applications of electrically switchable CLC structures.

Ÿ   The movement and disappearance of dislocations during the relaxation process, which are related to the number of layers in the CLC, are intriguing. Could the authors provide more details on how the pitch (p) or d/p ratio changes during the relaxation process?

Ÿ   Did the authors observe any flows in the CLC system during the relaxation processes of the electrically induced structural transformations? If so, could they provide a characterization of these flows?

Ÿ   There are typographical errors in lines 72 and 74 of the manuscript. Please correct them.

Ÿ   Incomplete theoretical explanations: (1) the reason behind the transition to an azimuthally uniform state after applying the electric field and allowing for static relaxation; (2) the mechanism underlying the change in the azimuthal orientation angle of the liquid crystal director upon applying the electric field.

Ÿ   Insufficient conclusion section: While the manuscript focuses on the differences in the electro-optic response and relaxation processes between the two orienting polymer mixtures, further discussion on the advantages, disadvantages, and potential applications of each mixture would enhance the value of the manuscript. Readers would be interested in a systematic discussion and outlook, such as the performance of similar compounds, the potential for further research on memory-like structures, and the controllability of the formed defect lines.

Ÿ   In Figure 1, there seems to be a typographical error where 'd' is mistakenly written as 'c'.

Ÿ   In Figure 3, the decimal points for 1.3V and 1.4V could be expressed using a period instead of a comma to ensure clarity. These are minor issues in the writing.

Comments on the Quality of English Language

Ÿ   Please revise and proofread the manuscript for clarity and accuracy.

Reviewer 2 Report

Comments and Suggestions for Authors

This manuscript studied the electrically induced structural transformations within CLC cells with tangential-conical boundary conditions at the various director tilt angles on the substrate under conical anchoring. The composition influence of the orienting polymer films on the director tilt angles, the formation of orientational structures in the LC layer, as well as the electro-optic response and relaxation processes are considered. The manuscript is recommended for publication in this journal after addressing the following issues:

1.  There are some careless mistakes in the manuscript. Such as in caption of Figure 1, “U = 10 V, and (c), (e) 1 min, (f) 120 min, (g) 300 min after voltage-off” should be corrected to “U = 10 V, and (d), (e) 1 min, (f) 120 min, (g) 300 min after voltage-off”.

2.   It’s recommended to add the angle between analyzer and polarizer in Figure 3 as there are serval kinds of double arrows appeared at the first time in the manuscript.

Round 2

Reviewer 1 Report

Comments and Suggestions for Authors

no comment